# UniIF: Unified Molecule Inverse Folding

**Zhangyang Gao** [1,2,†], **Jue Wang** [1,2,†], **Cheng Tan** [1,2,†],
**Lirong Wu** [2], **Yufei Huang** [2], **Siyuan Li** [2], **Zhirui Ye** [2], **Stan Z. Li** [2,*]

[1] Zhejiang University [2] Westlake University

## Abstract

Molecule inverse folding has been a long-standing challenge in chemistry and biology, with the potential to revolutionize drug discovery and material science. Despite specified models have been proposed for different small- or macro-molecules, few have attempted to unify the learning process, resulting in redundant efforts. Complementary to recent advancements in molecular structure prediction, such as RoseTTAFold All-Atom and AlphaFold3, we propose the unified model UniIF for the inverse folding of all molecules. We do such unification in two levels: 1) Data-Level: We propose a unified block graph data form for all molecules, including the local frame building and geometric feature initialization. 2) Model-Level: We introduce a geometric block attention network, comprising a geometric interaction, interactive attention and virtual long-term dependency modules, to capture the 3D interactions of all molecules. Through comprehensive evaluations across various tasks such as protein design, RNA design, and material design, we demonstrate that our proposed method surpasses state-of-the-art methods on all tasks. UniIF offers a versatile and effective solution for general molecule inverse folding.

## 1 Introduction

Molecule inverse folding plays a pivotal role in drug and material design, enabling scientists to synthesize novel molecules with the desired structure. Previously, many studies focus on either macromolecules [19, 33, 21, 9, 17, 4, 10, 17, 8, 34] or small molecules [6, 26, 16, 28, 14, 31] separately, leaving the challenge of inverse folding general molecules. For example, the advanced small molecule model [6, 31] take atoms as basic units; the macromolecule models [8, 10] consider predefined microstructures (such as amino acids and nucleotides) as the basic units. Additionally, even for the same molecule, different models employ varying strategies to extract geometric features. Complementary to the great success of RoseTTAFold All-Atom [25] and AlphaFold3 [1] in molecular structure prediction, we propose a unified model, UniIF, for the inverse folding of all molecules.

By comparing small- and macro-molecules, we identify three challenges toward the unified model: (1) **Unit Discrepancy**: The macromolecules takes predefined microstructures (amino acids and nucleotides) as the basic units, while small molecules takes atoms as basic units. (2) **Geometric Featurizer**: Different studies employ various strategies for extracting geometric features from structures, such as distance, angles and tensor product; there are lack of unified featurization strategy. (3) **System Size**: The small-molecules allow the full attention transformer to learn long-term dependencies, but the quadratic computing cost limits the mechanism scalling up to macro-molecular systems. Alternatively, previous research use sparse

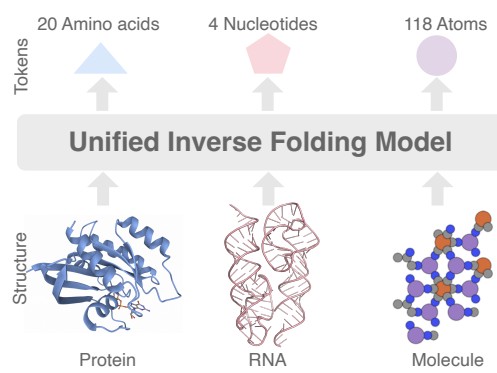

**Figure 1:** Unified molecule inverse folding.

GNN, which suffers from the limited local receptive field that causes over-smoothing and over-squashing [29]. In addition, developing a unified model working well for all molecules is challenging.

---

[†]Equal Contribution, [*]Corresponding Author.

38th Conference on Neural Information Processing Systems (NeurIPS 2024).

The unit discrepancy makes it challenge to adapt methods across small- and macro-molecules, which explains the divergence between these two research lines. As a solution, we propose a frame-based block to unify the representation of amino acids, nucleotides, and atoms: a group of atoms with varying size is treated as a block with fixed size. Each block includes decoupled equivariant basis and invariant features, generalizing the representation of AlphaFold2 and other small-molecule methods.

The geometric featurizer is necessary to capture the geometric interactions between blocks. We initialize the equivariant block basis using predefined rules or a learnable GNN layer, and then constructing invariant block features based on these basis. The key operation is to use local coordinates and dot product to capture the geometric interactions between virtual atoms. We reuse the featurizer in each model layer to interactively learn updated geometric features, where the concept of directed virtual atom is introduced to enhance the pairwise interactions. We show that the unified featurizer works well across protein design, RNA design, and material design.

We use sparse GNN to address the system size issue, while maintaining the ability to capture long-term dependencies. The transformer-style protein models like AlphaFold and RosettaFold require a substantial amount of GPU memory. Sparse GNNs, on the other hand, are criticized for their tendency to over-smooth and over-squash due to their limited local receptive field. To be efficient while preserving the ability to capture long-term dependencies, we introduce global virtual blocks. Each virtual block is connected to all real blocks, serving as an information exchange agent.

We conducted comprehensive experiments across various tasks, including protein design, RNA design, and material design, to demonstrate the effectiveness of UniIF. The results show that UniIF achieves state-of-the-art performance on all the tasks, which is non-trivial and may benefit the machine learning, drug discovery, and material science communities.

## 2 Related work

**Unification.** Unified molecular learning has attracted increasing attention in recent years. RoseTTAFold All-Atom (RFAA) [25] and AlphaFold3 [1] are two representative models that have achieved remarkable success in protein structure prediction. RoseTTAFold All-Atom uses an atom-bond graph for small molecules and a frame graph for macromolecules. AlphaFold3 uses a bi-level representations, i.e., atom representation and token representation, for all molecules. The token concept is requivalent to the block concept in this paper, which means a group of atoms, such as a amino acid or a nucleotide. GET [24] and EPT [20] are two recent models that use a block representation for both small and macromolecules and introduce a new equivariant transformer backbone. Unlike RFAA [25], which specifies a atom-bond graph for small molecules, our model employs a unified block graph for all molecule types and do not require the atom-bond graph. Our model also differs from AlphaFold3 [1], GET [24] and EPT [20] in the that we introduce the vector basis for each block.

**Protein Inverse Folding.** Recent research use $k$-NN graph to represent the 3D structure and employ graph neural networks for protein inverse folding. GraphTrans [19] uses the graph attention encoder and autoregressive decoder for protein design. GVP [21] proposes geometric vector perceptrons to learn from both scalar and vector features. GCA [33] introduces global graph attention for learning contextual features. In addition, ProteinSolver [32] is developed for scenarios where partial sequences are known while not reporting results on standard benchmarks. Recently, AlphaDesign [9], ProteinMPNN [4], ESMIF [17], LMDesign [40], KWDesign [8], VFN [27] achieves dramatic improvements. A benchmark [11] is proposed to comprehensively evaluate protein design models.

**RNA Inverse Folding.** RNA inverse folding is a challenging task due to the complex secondary structure and tertiary structure. Traditional methods [3] include colony optimization and constraint programming, in addition to adaptive walk, simulated annealing and Boltzmann sampling. Recent deep learning method RDesign [34] has achieved promising results and build a benchmark for AI researchers to follow-up. RiboDiffusion [18] use diffusion decoder to generate RNA sequences conditioned on the backbone structure embeddings.

**Material Design.** Deciding which chemical compositions are likely to form compounds is a critical task in material design [28, 26, 14]. An important application is to substitute lattice-site elements or ionic species within existing compounds that exhibit similar chemical behaviors. Wang et al. [36] successfully employed the elemental substitution method to discover 18,479 stable compounds out of a pool of 189,981 potential candidates. Recently, Jensen et al. [6] introduced a open dataset for material design and established a benchmark for evaluating deep learning models in this domain.

# 3 Method

## 3.1 Overall Framework

As shown in Fig. 2, we propose the unified model for general molecule inverse folding. The key insights include: (1) transforming all molecules into block graphs, where each block represents an amino acid, nucleotide, or atom; (2) proposing a geometric featurizer to initialize geometric node and edge features; and (3) introducing a new GNN layer with long-term dependencies to learn expressive block representations. Our unified model achieves competitive results across diverse tasks, including protein design, RNA design, and material design.

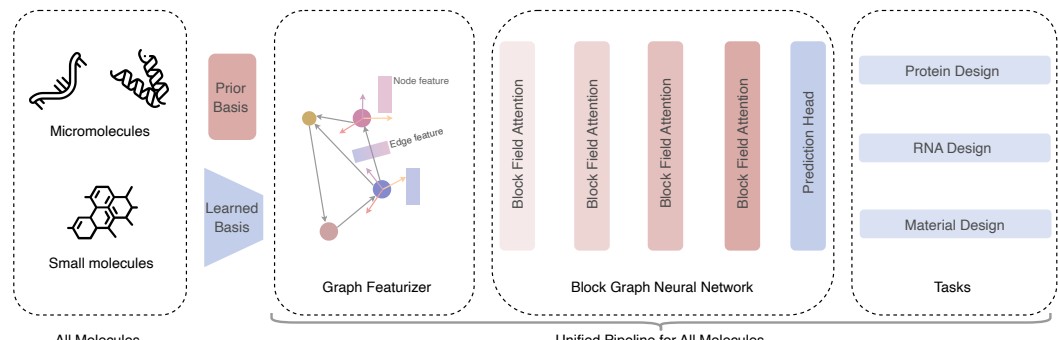

**Figure 2:** The Overall framework. (1) The model treat all types of molecules as block graphs. For macromolecules, we use predefined frames based on amino acids and nucleotides; for small molecules, we learn the local frame of each block by one-layer GNN. (2) A geometric featurizer is used to initialize the geometric node feature and edge features. (3) We propose the block graph attention layer, based on which we build the block graph neural network to learn expressive block representations. (4) Finally, we show that the UniIF can achieve competitive results on diverse tasks, ranging from protein design, RNA design and material design.

## 3.2 Block Graph

We introduce the block graph to represent all types of molecules, where the key insight is to transform irregular set of atoms (varying size) as regular block representation (fixed size).

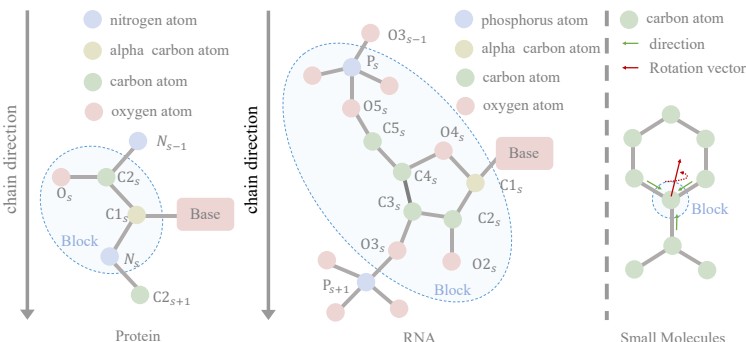

**Figure 3:** Blocks of different molecules. The basic building blocks include amino acids, nucleotides and atoms.

**Atom-based Block Representation.** A block $\mathcal{B} = \{(\boldsymbol{x}_i, \boldsymbol{z}_i)\}_{i=1}^{|\mathcal{B}|}$ contains a set of atoms $\{\boldsymbol{x}_i, \boldsymbol{z}_i\}_{i=1}^{|\mathcal{B}|}$, where $\boldsymbol{x}_i \in \mathbb{R}^3$ and $\boldsymbol{z}_i \in \mathbb{R}^d$ represent the equivariant coordinate and invariant features, such as the atom type. The common block types include amino acids, nucleotides, and atoms, which are represented as $\mathcal{B}^{fold}$, $\mathcal{B}^{rna}$, and $\mathcal{B}^{smol}$, respectively. Formally, we write $\mathcal{B}^{fold} = \{(\boldsymbol{x}_i, \boldsymbol{z}_i)\}_{i \in \mathcal{V}^{fold}}$ and $\mathcal{B}^{rna} = \{(\boldsymbol{x}_i, \boldsymbol{z}_i)\}_{i \in \mathcal{V}^{rna}}$, where $\mathcal{V}^{fold} = \{\text{N}, \text{C1}, \text{C2}, \text{O}\}$ and $\mathcal{V}^{rna} = \{\text{P}, \text{C5}, \text{C4}, \text{C3}, \text{C2}, \text{C1}, \text{O5}, \text{O4}, \text{O3}, \text{O2}\}$ are the sets of atoms for amino acids and nucleotides, respectively. For small molecules, each atom $a$ represents a block $\mathcal{B}^{smol} = \{(\boldsymbol{x}_a, \boldsymbol{z}_a)\}$, where $1 \leq a \leq 118$. As the block size $|\mathcal{B}|$ varies for different types of blocks, the atom-based blocks representation could not directly be applied for unified modeling.

**Frame-based Block Representation.** We introduce frame-based block representation to unify the modeling of all molecules. A block $\mathcal{B} = (F, \boldsymbol{f})$ contains the equivariant frame $F$ and invariant feature vector $\boldsymbol{f} \in \mathbb{R}^d$. The local frame $F(R, \boldsymbol{t})$ contains the axis matrix $R = [\boldsymbol{e}_1, \boldsymbol{e}_2, \boldsymbol{e}_3, \cdots, \boldsymbol{e}_u]$ and translation vector $\boldsymbol{t}$. We set $\boldsymbol{t}$ as the coordinate of the representative atom, i.e., $C1$ of macromolecules and the atom itself of small molecules. Following AlphaFold2, we consider the special case that $R \in \mathbb{R}^{3,3}$ is orthogonal. However, additional experiments show that the model can also work well with non-orthogonal axis matrix. For macromolecules, the axis matrix $R$ is predefined based on amino acids and nucleotides, while for small molecules,

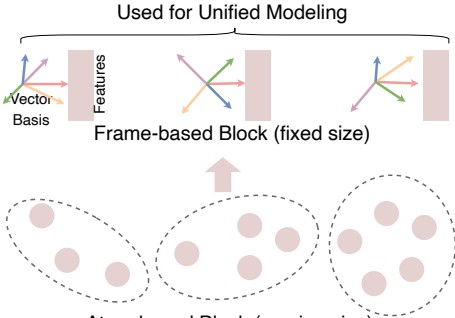

Frame-based Block (fixed size)

Atom-based Block (varying size)
**Figure 4:** Unified molecule inverse folding.

we learn the axis matrix $R$ as it does not have prior common structure patterns. The frame-based block representation decouples geometric information: (1) the local frame basis describe the equivariant pose; (2) the invariant feature vector could embed the atom type and invariant local structure patterns for different tasks. More importantly, the dimension of the frame-based block representation is fixed, which is beneficial for the unified modeling; we build block features in Sec. 3.3.

**Frame-based Block Graph.** Given a molecule $\mathcal{M} = \{\mathcal{B}_s\}_{s=1}^{n}$ containing $n$ blocks, we build the block graph $\mathcal{G}(\{\mathcal{B}_s\}_{s=1}^{n}, \mathcal{E})$ using kNN algorithm. In the block graph, the $s$-th node is represented as $\mathcal{B}_s = (F_s, \boldsymbol{f}_s)$, and the edge between $(s, t)$ is represented as $\mathcal{B}_{st} = (F_{st}, \boldsymbol{f}_{st})$. The relative frame is defined as $F_{st} = F_s^{-1} \circ F_t$. Inspired by [22, 10, 27], we modify PiFold featurizer to initialize the geometric node feature $\boldsymbol{f}_s$ and edge feature $\boldsymbol{f}_{s,t}$; refer to Sec. 3.3.

**Relation to Other Methods.** The frame-based block is a generalized data form of AlphaFold2 and other methods. If $R$ is required to be a rotation matrix, the frame-based block is equivalent to AlphaFold2's local frame; otherwise, it is equivalent to represent the atom as invariant feature $\boldsymbol{h}$ and equivalent vector $\boldsymbol{x}$, similar to GVP [21] and DimeNet [12].

### 3.3 Block Graph Featurizer

**Learning Local Frame.** For small molecules, there is no predefined local frame, and we need to learn the local frame for each atom. Given the the molecule $\mathcal{M} = \{(\boldsymbol{x}_s, \boldsymbol{z}_s)\}_{s=1}^{|\mathcal{M}|}$, we use a 1-layer of GNN to initialize the atom representation $\{\boldsymbol{z}_s\}_{s=1}^{|\mathcal{M}|} \leftarrow \texttt{BlockGAT}(\{\boldsymbol{x}_s, \boldsymbol{z}_s\}_{s=1}^{|\mathcal{M}|})$, where the inital local frames are $T(\mathbf{I}, \boldsymbol{0})$. The rotation vector $\boldsymbol{r}_s$ of the $s$-th atom is constructed by message passing:

$$\boldsymbol{r}_s = (r_x, r_y, r_z) = \sum_{k \in \mathcal{N}_s} \frac{e^{\texttt{MLP}(\boldsymbol{z}_s, \boldsymbol{z}_t)}}{\sum_{k \in \mathcal{N}_s} e^{\texttt{MLP}(\boldsymbol{z}_s, \boldsymbol{z}_k)}} \frac{\boldsymbol{x}_k - \boldsymbol{x}_s}{||\boldsymbol{x}_k - \boldsymbol{x}_s||} \quad (1)$$

$\mathcal{N}_s$ is the $s$-th atom's neighbor system. Let the direction $(r_x, r_y, r_z) = \frac{\boldsymbol{r}_s}{||\boldsymbol{r}_s||}$ and magnitude $\theta = ||\boldsymbol{r}_s||$ represent the rotation axis and angle, we compute the quanternion $\boldsymbol{q}_s$ and rotation matrix $R_s$:

$$\begin{cases} \boldsymbol{q}_s = [w, x, y, z] = [\cos \frac{\theta}{2}, r_x \sin \frac{\theta}{2}, r_y \sin \frac{\theta}{2}, r_z \sin \frac{\theta}{2}] \\ R_s = [\boldsymbol{e}_x, \boldsymbol{e}_y, \boldsymbol{e}_z] = \begin{bmatrix} 1 - 2y^2 - 2z^2 & 2xy - 2zw & 2xz + 2yw \\ 2xy + 2zw & 1 - 2x^2 - 2z^2 & 2yz - 2xw \\ 2xz - 2yw & 2yz + 2xw & 1 - 2x^2 - 2y^2 \end{bmatrix} \end{cases} \quad (2)$$

Finally, the local frame of the $s$-th atom is $T_s(R_s, \boldsymbol{t}_s)$, where $\boldsymbol{t}_s$ is the atom coordinate. Experiments show that learning rotation vectors consistently outperforms learning Schmidt-orthogonalized axises.

**Node Geometric Feature.** The invariant block feature captures the atom type and the local structure:

$$\begin{cases} \boldsymbol{z}_i^{pos} = R_s^T(\boldsymbol{x}_i - \boldsymbol{t}_s) = F_s^{-1} \circ \boldsymbol{x}_i & \text{Equivalent to invariant features, local structure} \\ \boldsymbol{f}_s = \frac{1}{|\mathcal{B}_s|} \sum_{i \in \mathcal{B}_s} \texttt{MLP}(\boldsymbol{z}_i, \boldsymbol{z}_i^{pos}) & \text{Pooling atom features as block features, embed atom type} \end{cases} \quad (3)$$

The inverse frame operation $T_s^{-1}$ project the equivalent global coordinates to the invariant local coordinate, i.e., $\boldsymbol{x}^{local} = F_s^{-1} \circ \boldsymbol{x}^{global} = R_s^T(\boldsymbol{x}^{global} - \boldsymbol{t}_s)$. We use MLP to embed atom type and local coordinates. All atom features in the same block are pooled to get the block feature $\boldsymbol{f}_s$.

**Edge Geometric Feature.**    We initialize pairwise features following the principle that

*Edge features capture the directed 3D interactions.*

Instead of using mutually constructed distance and angle features, we concatenate the local coordinates of two blocks to fully describe their 3D positions. Given $\mathcal{B}_s$ and $\mathcal{B}_t$ with global coordinate matrices as $X_s \in \mathbb{R}^{|v_s|,3}$ and $X_t \in \mathbb{R}^{|v_t|,3}$, the invariant edge features following $s \leftarrow t$ direction is

$$\boldsymbol{f}_{s,t} = T_s^{-1} \circ ([X_s \| X_t]) \tag{4}$$

where $T_s^{-1} = (R_s^T, -R_s^T \boldsymbol{t}_s)$ projects equivariant global coordinates to invariant local coordinates.

### 3.4 Block Graph Attention Module

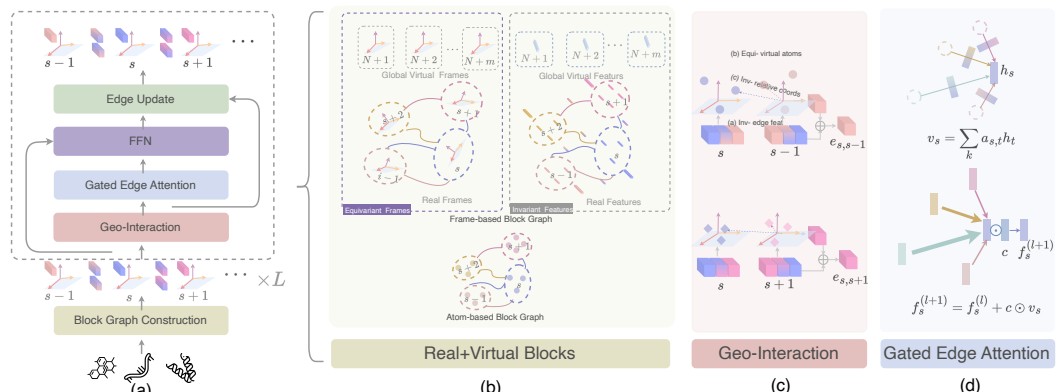

**Figure 5:** Block Graph Attention Module. (a) Virtual Block for Long-term Dependencies. (b) Geometric Interaction Extractor for learning pairwise features. (c) Gated Edge Attention for updating node features.

**Frame-based SE-(3) Module Design.**    Given the geometric transformation $\boldsymbol{y} = W\boldsymbol{x}$, we decompose $W = R_t \Sigma R_s^T$ using SVD and explain $\boldsymbol{y} = R_t \Sigma R_s^T \boldsymbol{x}$ as:

1. Projecting $\boldsymbol{x}$ as the local coordinate $\boldsymbol{x}^{local}$ using the frame $F_s(R_s, \boldsymbol{0})$, i.e., $\boldsymbol{x}^{local} = R_s^T \boldsymbol{x}$.

2. Updating local coordinates via gated attention, i.e., $\boldsymbol{x}^{local} \leftarrow \Sigma \boldsymbol{x}^{local}$.

3. Translating $\boldsymbol{x}^{local}$ as the global coordinate using frame $T_t(R_t, \boldsymbol{0})$, i.e., $\boldsymbol{y} = R_t \boldsymbol{x}^{local}$.

If we parameterize $\Sigma \boldsymbol{y}$ as $f_\theta(\boldsymbol{y})$, and considers the effects of translation, i.e., $T_s(R_s, \boldsymbol{t}_s), T_t(R_t, \boldsymbol{t}_t)$, the general principle of designing SE-(3) networks could be:

$$\begin{cases} \hat{\boldsymbol{z}} = R_s^{-1}(\boldsymbol{x} - \boldsymbol{t}_s) = T_s^{-1} \circ \boldsymbol{x} & \text{Equivalent to invariant} \\ \hat{\boldsymbol{z}} \leftarrow f_\theta(\hat{\boldsymbol{z}}) & \text{Invariant update, one can use GNN or Transformer} \\ \hat{\boldsymbol{x}} = R_t \hat{\boldsymbol{y}} + \boldsymbol{t}_t = T_t \circ \hat{\boldsymbol{z}} & \text{Invariant to equivalent} \end{cases} \tag{5}$$

The well-known AlphaFold actually follows such a design, where they parameterize $f_\theta$ as the IPA module. In this work, we replace $f_\theta$ with an enhanced graph neural network:

$$\boldsymbol{f}_s^{(l+1)}, \boldsymbol{f}_{st}^{(l+1)} \leftarrow f_\theta(\boldsymbol{f}_s^{(l)}, \boldsymbol{f}_{st}^{(l)} | T_s, T_{st}, \mathcal{E}) \tag{6}$$

where $\boldsymbol{f}_s^{(l)}$ and $\boldsymbol{f}_{st}^{(l)}$ represent the input node and edge features of the $l$-th layer. In Fig. 5, we show the design of the Block Graph Attention Module, consisting of three components: (1) geometric interaction extractor, (2) virtual block for long-term dependencies, and (3) the edge attention mechanism. We show the detailed design of the Block Graph Attention Module in the following sections.

**Long-term Dependency via Virtual Blocks.** The GNN is criticized by local receptive field, yielding the problem of over-smoothing and over-squashing [2, 5, 29, 13]. Transformers overcome these problems using direct paths between distant nodes, while suffering from the $\mathcal{O}(n^2)$ computing cost. We introduce $n'$ virtual blocks $\{\mathcal{B}_i\}_{i=n}^{n+n'}$ as information agents for a graph. Each virtual block directly connects to all the real blocks, resulting in $(2 \cdot n \cdot n')$ additional directed edges. As $n' \ll n$, we claim that the computing cost is close to original GNN. All the virtual blocks, i.e., $T_{n+1}(R', \boldsymbol{t}'), T_{n+2}(R', \boldsymbol{t}'), \cdots, T_{n+n'}(R', \boldsymbol{t}')$, share the same rotation $R'$ and translation $\boldsymbol{t}'$:

$$\begin{cases} \mathbf{X} = [\boldsymbol{x}_1, \boldsymbol{x}_2, \cdots] & \text{All the coordinates of the molecule} \\ \mathbf{X}^T\mathbf{X} = U\Lambda V^T & \text{SVD} \\ R' = UV^T = [\boldsymbol{e}_x, \boldsymbol{e}_y, \boldsymbol{e}_z] \\ \boldsymbol{t}' = \frac{\sum_{i=1}^N \boldsymbol{x}_i}{N} & \text{Center of Mass} \end{cases} \tag{7}$$

The invariant features of virtual blocks are different, as we hope they learn diverse interactions. We encode the index to initialize block features $\boldsymbol{f}_i' = \texttt{Embedding}(i)$ for $i \in \{1, 2, \cdots, n'\}$.

**Geometric Interaction Extractor.** We enhance edge features with geometric interactions using the local coordinates of virtual inter-atoms and dot products of virtual intra-atoms. Previous works, such as PiFold [10], introduced virtual atoms in the featurizer to capture informative side-chain geometry beyond protein backbones, resulting in performance gains. VFN [27] extended this idea by allowing GNN layers to update the virtual atoms. However, these efforts are limited to learning virtual intra-atoms conditioned on node features. Instead, we propose virtual inter-atoms conditioned

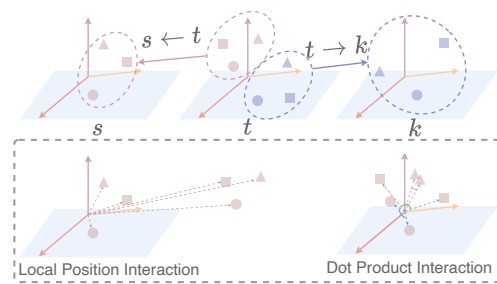

**Figure 6:** Geometric Interactions.

on edge features, allowing the same node to exhibit different virtual states specified by edges. Additionally, inspired by small molecule modeling, we use the dot product of virtual intra-atoms to capture angle information. We show the geometric interactions in Fig. 6, and formulate it as:

$$\begin{cases} \boldsymbol{h}_{st}^{(l)}, \boldsymbol{h}_t^{(l)} = \texttt{MLP}(\boldsymbol{f}_{st}^{(l)}), \texttt{MLP}(\boldsymbol{f}_t^{(l)}) \in \mathbb{R}^{m,3} & \text{Edge feature} \\ \hat{\boldsymbol{z}}_{st}^{(l)} = (T_{st} \circ \boldsymbol{h}_{st}^{(l)}) \| \boldsymbol{h}_{st}^{(l)} & \text{Local coordinates of virtual inter-atoms} \\ \boldsymbol{a}_{st} = \boldsymbol{h}_s^T R_s^T R_t \boldsymbol{h}_t & \text{Geometric dot product of virtual intra-atoms} \\ \boldsymbol{g}_{st}^{(l)} = \texttt{MLP}(\hat{\boldsymbol{z}}_{st}^{(l)}, q_{st}, r_{st}, \boldsymbol{a}_{st}) \\ \boldsymbol{f}_{st}^{(l)} \leftarrow \texttt{MLP}(\boldsymbol{f}_{st}^{(l)}, \boldsymbol{g}_{st}^{(l)}) \end{cases} \tag{8}$$

where $T_{st} = T_s^{-1} \circ T_t = (R_s^{-1}R_t, R_s^{-1}(\boldsymbol{t}_t - \boldsymbol{t}_s))$, $q_{st} = \text{vec}(R_{st}) \in \mathbb{R}^9$ is the flatten rotation matrix of $T_{st}$, and $r_{st} = \|\boldsymbol{t}_s - \boldsymbol{t}_t\|$ indicates the pairwise distance. All $q_{st}, r_{st}$ an $\boldsymbol{a}_{st}$ are invariant features. We highlight the difference to previous researches in color.

**Gated Edge Attention.** We modify PiFold's GNN to capture the geometric interactions when updating node features. For molecular design tasks, we find that aggregating edge features only leads to consistent performance gains. We understand this phenomenon as the model can pay more attention on learning 3D interactions under such a model design. In addition, we use a gated mechanism to control how the edge features are injected to node features. The gated edge attention module is:

$$\begin{cases} w_{st} = \texttt{AttMLP}(\boldsymbol{f}_s^{(l)} \| \boldsymbol{f}_{st}^{(l)} \| \boldsymbol{f}_t^{(l)}) \\ a_{st} = \frac{\exp w_{st}}{\sum_{k \in \mathcal{N}_s} \exp w_{sk}} \\ \boldsymbol{h}_t^{(l)} = \texttt{EdgeMLP}(\boldsymbol{f}_{st}^{(l)}) & \text{Only conditiond on edge} \\ \Delta \boldsymbol{f}_s^{(l)} = \sum_{t \in \mathcal{N}_s} a_{st} \boldsymbol{h}_t^{(l)} \\ \boldsymbol{f}_s^{(l+1)} = \boldsymbol{f}_s^{(l)} + \sigma(\texttt{MLP}(\Delta \boldsymbol{f}_s^{(l)})) \odot \Delta \boldsymbol{f}_s^{(l)} & \text{Update node feature via forget gate} \end{cases} \tag{9}$$

where $\odot$ is element-wise product operation, and $\sigma(\cdot)$ is the sigmoid function. We highlight the difference between PiGNN and the proposed module in color.

**FFN & Edge Updating.** Analogous to the transformer model, the FFN is a MLP. The edge updating layer remains the same as PiFold:

$$\boldsymbol{f}_{st} = \texttt{EdgeMLP}(\boldsymbol{f}_s || \boldsymbol{f}_{st} || \boldsymbol{f}_t) \tag{10}$$

The proposed module could be equivalently implemented as a transformer module using matrix multiplication. However, we find that padding proteins to the maximum length would greatly increase the computing cost in both GPU occupancy and runtime; We suggest using GNN without padding.

**Regularization.** We find that the proposed model fit training data better than PiFold and more likely to suffer from overfitting. To address this issue, we randomly drop out the nodes/edges with a probability of $p$ to prevent overfitting. We find that controlling the dropout rate could result in models with different fitting abilities. The best performance is achieved when $p = 0.05$.

## 4 Experiments

We show the effectiveness of UniIF via multiple inverse folding tasks and ablation studies. We briefly introduce molecular design tasks as follows:

- **Protein Design (T1):** Designing protein sequences folding into the target structure.
- **RNA Design (T2):** Designing RNA sequences folding into the target structure.
- **Material Design (T3):** Discovering stable composition from a known material structure.

### 4.1 Protein Design (T1)

**Task Description** Protein design aims to design protein sequences that fold into target structures. Given a protein backbone structure $\mathcal{X} = \{X_i \in \mathbb{R}^{m,3} : 1 \leq i \leq n\}$, where $m$ is the maximum number of points belonging to the $i$-th residue, $n$ is the number of residues and the natural proteins are composed by 20 types of amino acids, the goal is to learn a function $\mathcal{F}_\theta$:

$$\mathcal{F}_\theta : \mathcal{X} \mapsto \hat{\mathcal{S}}. \tag{11}$$

The parameters $\theta$ are learned by minimizing the cross-entropy loss, i.e., $\mathcal{L}(\mathcal{F}_\theta(\mathcal{X}), \mathcal{S}) = -\sum_{i=1}^{n} \log s_i p(\hat{s}_i | \mathcal{X}, \theta)$. The task is challenging due to the combinatorial search space of amino acids and the complex relationship between sequence and structure.

**Settings** We evaluate of UniIF on the CATH4.3 dataset [30] following prior works [11, 8]. The dataset is split by the CATH topology classification code, yielding 16,631 training, 1,516 validation, and 1,864 testing samples. To assess generalization, we adopt a time-split strategy, considering the use of pretrained ESM2 models by some baselines, which risk data leakage. The time-split evaluation assigns data before a specific date to the training set and data after that date to the test set. For structural time-split evaluation, we use the CASP15 dataset [11], containing novel crystal structures not seen during training. For sequence time-split evaluation, we use the NovelPro dataset [8], which includes 76 protein sequences released within 30 days before November 23, 2023, with structures predicted by AlphaFold2. UniIF consists of 10 layers of BlockGAT with a hidden dimension of 128. It is trained using the Adam optimizer with a learning rate of 1e-3 and a batch size of 8 for 50 epochs.

**Metrics & Baselines** We report the median recovery rate of the top-1 predicted sequences, representing the percentage of correctly predicted residues. The ESM2-free baselines include StructGNN [19], GraphTrans [19], GCA [33], GVP [21], AlphaDesign [9], ProteinMPNN [4], and PiFold [10]. The ESM2-based baselines include LMDesign [17] and KWDesign [8]. While we prefer open-source baselines, we also re-implement VFN [27] for a comprehensive comparison.

**Conclusion** We provide results under different settings (with and without ESM2) and across diverse datasets (CATH4.3, CASP, NovelPro). Using a pure inverse folding model without ESM2, UniIF achieves the best performance on all datasets, demonstrating its effectiveness. Notably, UniIF outperforms the strong baseline PiFold with fewer learnable parameters. In time-split evaluations, UniIF surpasses all baselines, including ESM2-based methods, by a significant margin. On NovelPro, which features novel sequences, UniIF outperforms LMDesign and KWDesign that use ESM2 for sequence refinement. This indicates UniIF's superior generalizability, crucial for real-world

| w ESM | Model length | Rec % ↑ (CATH4.3) | | | | Rec % ↑ CASP | Rec % ↑ NovelPro | Params |
|---|---|---|---|---|---|---|---|---|
| | | $L < 100$ | $100 \leq L < 300$ | $300 \leq L < 500$ | Full | | | |
| ✓ | LMDesign [17] | 0.47 | 0.56 | 0.61 | 0.56 | 0.48 | 0.59 | |
| | KWDesign [8] | 0.51 | 0.61 | 0.69 | 0.60 | 0.56 | 0.64 | |
| ✗ | StructGNN [19] | 0.30 | 0.34 | 0.40 | 0.34 | 0.36 | 0.40 | 1.4M |
| | GraphTrans [19] | 0.29 | 0.34 | 0.39 | 0.34 | 0.35 | 0.40 | 1.5M |
| | GCA [33] | 0.32 | 0.36 | 0.41 | 0.36 | 0.40 | 0.43 | 2.1M |
| | GVP [21] | 0.33 | 0.38 | 0.45 | 0.38 | 0.39 | 0.42 | 0.9M |
| | AlphaDesign [9] | 0.37 | 0.43 | 0.47 | 0.42 | 0.42 | 0.46 | 3.6M |
| | ProteinMPNN [4] | 0.38 | 0.44 | 0.52 | 0.44 | 0.44 | 0.52 | 1.7M |
| | PiFold [10] | 0.43 | 0.52 | 0.59 | 0.51 | 0.47 | 0.57 | 5.8M |
| | UniIF (ours) | **0.45** | **0.54** | **0.61** | **0.53** | **0.51** | **0.66** | 5.4M |
| Ablation | VFN [27] | 0.45 | 0.53 | 0.60 | 0.52 | 0.48 | 0.63 | 5.4M |
| | -GDP | 0.45 | 0.53 | 0.61 | 0.52 | 0.50 | 0.65 | 5.3M |
| | -EAttn | 0.44 | 0.53 | 0.60 | 0.52 | 0.48 | 0.63 | 5.7M |
| | -VFrame | 0.45 | 0.53 | 0.61 | 0.52 | 0.49 | 0.64 | 5.4M |

**Table 1:** Protein Design results. The **best** and suboptimal results are labeled with bold and underlined. "VFN" means that we replace the geometric interaction operation with VFN's operation [27]. "-GDP" means that we remove the geometric dot product features. "-EAttn" means that we replace the gated edge attention with PiGNN's attention module [10]. "-VFrame" means that we remove the global virtual frames.

applications. Ablation studies show that the proposed geometric featurizer, gated edge attention, and global virtual frame enhance performance. On CATH4.3, the overall improvement is slight due to strong baselines, but time-split evaluation highlights UniIF's superiority in generalization.

## 4.2 RNA Design (T2)

**Task Description**   Similar to protein design, RNA design aims to design RNA sequences that fold into target structures. Specially, previous work [34] use the RNA secondary structure as additional input to guide the design process, since the tertiary structure is limited. In this work, we only use the tertiary structures as input for the reason of unification, which is more challenging than the baselines.

**Datasets & Baselines**   We conduct experiments RNA on the dataset collected by RDesign [34], consisting of 2218 RNA tertiary structures, which are divided into training (1774 structures), testing (223 structures), and validation (221 structures) sets based on their structural similarity. Following RDesign's benchmark, baseline methods include SeqRNN, SeqLSTM, StructMLP, StructGNN, and StructGNN, GraphTrans [19], PiFold [10] and RDesign [34]. Given the small number of data samples, we report the median recovery and its standard deviation for threee independent runs.

**Table 2:** The recovery of RNA design. The **best** and suboptimal results are labeled with bold and underlined.

| Method | Recovery (%) ↑ | | | |
|---|---|---|---|---|
| | Short | Medium | Long | All |
| SeqRNN (h=128) | 26.52±1.07 | 24.86±0.82 | 27.31±0.41 | 26.23±0.87 |
| SeqRNN (h=256) | 27.61±1.85 | 27.16±0.63 | 28.71±0.14 | 28.24±0.46 |
| SeqLSTM (h=128) | 23.48±1.07 | 26.32±0.05 | 26.78±1.12 | 24.70±0.64 |
| SeqLSTM (h=256) | 25.00±0.00 | 26.89±0.35 | 28.55±0.13 | 26.93±0.93 |
| StructMLP | 25.72±0.51 | 25.03±1.39 | 25.38±1.89 | 25.35±0.25 |
| StructGNN | 27.55±0.94 | 28.78±0.87 | 28.23±1.95 | 28.23±0.71 |
| GraphTrans [19] | 26.15±0.93 | 23.78±1.11 | 23.80±1.69 | 24.73±0.93 |
| PiFold [10] | 24.81±2.01 | 25.90±1.56 | 23.55±4.13 | 24.48±1.13 |
| RDesign [34] | 37.22±1.14 | 44.89±1.67 | **43.06**±0.08 | 41.53±0.38 |
| UniIF (drop 0.05) | **48.21** ±0.95 | 49.66 ±1.28 | 37.29 ±0.17 | **48.94** ±0.37 |
| UniIF (drop 0.0) | 42.86 ±0.87 | 48.45 ±1.04 | 39.23 ±0.09 | 44.29 ±0.29 |
| UniIF (drop 0.1) | 45.21 ±0.98 | **51.70** ±1.26 | 40.30 ±0.14 | 46.00 ±0.38 |
| UniIF (drop 0.2) | 46.97 ±1.04 | 48.11 ±1.37 | 42.00 ±0.18 | 47.19 ±0.45 |

**Conclusion**   As shown in Table 2, UniIF achieves the best performance in all cases. The improvement is significant, as previous strong baselines like PiFold only excelled in protein design. To our knowledge, UniIF is the first model to achieve state-of-the-art performance in both protein and RNA design tasks, demonstrating its versatility and effectiveness. Compared to RDesign, which uses additional secondary structure features, UniIF relies solely on tertiary structure input and still performs better. UniIF successfully unifies the protein and RNA design processes, paving the way for a unified inverse folding model for protein-RNA complexes in future developments.

## 4.3 Material Design (T3)

**Task Description**   Discovering stable atom compositions from known material structures is crucial for new material discovery [28, 26, 14]. This task is challenging due to the large composition space and the lack of large-scale data. Thanks to recent benchmark efforts [6], we can evaluate the performance of UniIF on this novel task.

**Datasets & Baselines**   We evaluated UniIF on the CHILI-3K dataset [6], which consists of nanomaterial graphs derived from mono-metal oxides. The dataset includes 53 metallic elements and one non-metallic element (oxygen), comprising 3,180 graphs, 6,959,085 nodes, and 49,624,440 edges. Following the official benchmark, the dataset is randomly split into training (80%), validation (10%), and testing (10%) sets. Baselines include GCN [23], PMLP [39], GraphSAGE [15], GAT [35], GraphUNet [7], GIN [38], and EdgeCNN [37]. Experiments are repeated three times with different seeds, using early stopping with a patience of 50 epochs, and trained up to 1000 epochs.

| Method | Rec % ↑ |
|---|---|
| Random | 1.6±0.0 |
| GCN [23] | 49.6±0.1 |
| PMLP [39] | 46.1±0.0 |
| GraphSAGE [15] | 49.1±0.4 |
| GAT [35] | 46.1±0.0 |
| GraphUNet [7] | 55.2±7.9 |
| GIN [38] | 58.7±0.2 |
| EdgeCNN [37] | 63.2±0.9 |
| UniIF (ours) | **75.3**±1.2 |
| - frame | 54.9±2.8 |
| - quat | 65.2±3.9 |

**Table 3:** CHILI-3K Results.

**Conclusion**   In Table 3, UniIF outperforms all baselines by a large margin. Ablation studies demonstrate the crucial role of the learned local frame in enhancing interaction feature extraction. In addition, how to learn the local frame is also important. In the "- quat" ablation, we try to learn the x, y, and z axes with Householder orthogonalization directly, but found it less effective, with the recovery rate dropping from 75.3% to 65.2%. This highlights the value of the proposed local frame learning mechanism.

## 4.4 Case Study

In Fig. 7, we show the designed protein and RNA sequences. In addition, we use AlphaFold3 [1] to re-fold the designed sequences into structures. The ground truth (gray), PiFold (green), and UniIF (pink) structures are alinged and compared. We observe that UniIF improves both the recovery and RMSD of the designed protein and RNA, demonstrating its effectiveness in inverse folding tasks.

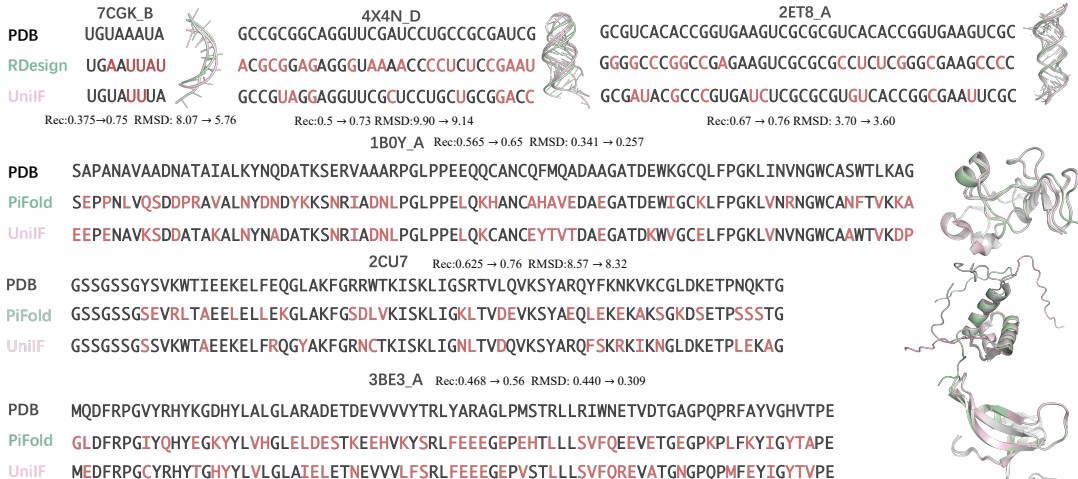

**Figure 7:** Designed examples. The ground truth (gray), PiFold (green), and UniIF (pink) structures are alinged.

## 5   Conclusion

We propose the first unified model, dubbled UniIF, for general molecule inverse folding. The key points include unifying the data representation, the featurizer and the model architecture without a drop in performance. Extensive experiments show that UniIF surpasses baseline methods on all tasks. Ablation studies reveal that the geometric interaction extractor, gated edge attention, and virtual long-term dependency modules contribute to performance gains. We believe that the proposed model can benefit multiple domains, such as machine learning, drug design, and material design.

## Acknowledgements

This work was supported by National Science and Technology Major Project (No. 2022ZD0115101), National Natural Science Foundation of China Project (No. U21A20427), Project (No. WU2022A009) from the Center of Synthetic Biology and Integrated Bioengineering of Westlake University and Integrated Bioengineering of Westlake University and Project (No. WU2023C019) from the Westlake University Industries of the Future Research Funding.

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
