# OpenReview forum: "UniIF: Unified Molecule Inverse Folding"
_NeurIPS.cc/2024/Conference — NeurIPS 2024 poster_

### Official Review · Reviewer_m3n9 · 2024-07-08

**Soundness:** 3
**Presentation:** 2
**Contribution:** 2
**Rating:** 5
**Confidence:** 1

**Summary:**

This paper proposes representing molecular systems as blocks of atoms and designs a neural network architecture that processes such representations. The method is applied for benchmark inverse folding tasks for protein/RNA. A materials-related task is also explored.

**Strengths:**

- The proposed method seems to perform well on various protein/RNA inverse folding benchmarks that focus on the recovery rate metric. (this reviewer is not experienced in protein/RNA design tasks, so it is hard for this reviewer to put the results in context.)

- The idea of representing clusters of atoms as blocks is an interesting approach and is potentially useful for representing molecular systems across different scales. while such ideas have already been explored in several previous works as the authors noted, there are new elements such as vector basis for blocks.

**Weaknesses:**

- The protein inverse folding ablation seems to show the various introduced techniques do not influence model performance much. The insights of the models seem obscured from this perspective, especially considering block-based representations and architectures have already been explored in previous works (GET/EPT). The RNA design task does not include ablation, and this reviewer is not convinced by the materials-related task (see below). It is rather unclear what a reader can take away from this work in terms what techniques actually contribute to improved performance.

- If "unified" is the key word of the method, perhaps the authors should include experiments where the model is trained on various data modalities and see if that improves performances on individual tasks.

- The materials sections are rather strange in this paper. Inverse folding for proteins/RNAs can be understood as inferring the sequence from a given structure. The materials task is not introduced in great detail. What is being predicted? How is it "inverse folding?"

- To follow up on that, the baselines that the proposed model is compared against are not what's considered state-of-the-art in materials representation. For materials, there are popular benchmarks such as Matbench (https://matbench.materialsproject.org/), JARVIS (https://pages.nist.gov/jarvis_leaderboard/), or Open catalyst (https://opencatalystproject.org/index.html), where results of SOTA models are readily available. This reviewer is not convinced by the materials-related results presented as significant details regarding the dataset/goals are missing, and the baselines are not really used for materials.

**Questions:**

- In the experiments, what exactly does with/without ESM mean? How is ESM used? Why can't the proposed method also become "with ESM"? Why is it valuable to consider "without ESM"? Are models "with ESM" more expensive or what?
- Can the author clarify the materials task and why it is inverse folding? Can the authors compare to materials-focused SOTA methods in the space?

**Limitations:**

The authors did not write about the limitations of the proposed method.

---

> ### Author Rebuttal · Authors · 2024-08-07
>
> **W1** The protein inverse folding ablation seems to show the various introduced techniques do not influence model performance much. The insights of the models seem obscured from this perspective, especially considering block-based representations and architectures have already been explored in previous works (GET/EPT). The RNA design task does not include ablation, and this reviewer is not convinced by the materials-related task (see below). It is rather unclear what a reader can take away from this work in terms what techniques actually contribute to improved performance.
> >> **Reply** As protein design is more mature than RNA design, we do ablation on it. However, the protein design performance has reached a bottleneck stage where any improvement is difficult. Things will be more clear on RNA dataset. Here we provide the ablation results on RNA dataset:
>
> | model      | short | medium | long  | all   |
> |------------|-------|--------|-------|-------|
> | w/o VFrame | 44.44 | 49.25  | 37.40 | 45.45 |
> | w/o EAttn  | 47.89 | 49.00  | 36.71 | 48.28 |
> | w/o GDP    | 46.88 | 48.00  | 37.19 | 47.06 |
> | UniIF      | 48.21 | 49.66  | 37.29 | 48.94 |
>
> In addition, we have evaluated the performance of GET on protein design, which performes very poor.
>
>
> **W2** If "unified" is the key word of the method, perhaps the authors should include experiments where the model is trained on various data modalities and see if that improves performances on individual tasks.
> >>**Reply** Such a protocol may improve performance of protein-RNA or protein-molecule interaction prediction. However, for the design task, we observe that training on single dataset lead to better results, as we do not consider the complex interactions.
>
> **W3** The materials sections are rather strange in this paper. Inverse folding for proteins/RNAs can be understood as inferring the sequence from a given structure. The materials task is not introduced in great detail. What is being predicted? How is it "inverse folding?"
> **Q2** Can the author clarify the materials task and why it is inverse folding?
> >>**Reply** This is a new task defined in  Chili[1], which gives the material structure and predict the composition of atoms types that fit to this structure.
>
> **W4** To follow up on that, the baselines that the proposed model is compared against are not what's considered state-of-the-art in materials representation. For materials, there are popular benchmarks such as Matbench (https://matbench.materialsproject.org/), JARVIS (https://pages.nist.gov/jarvis_leaderboard/), or Open catalyst (https://opencatalystproject.org/index.html), where results of SOTA models are readily available. This reviewer is not convinced by the materials-related results presented as significant details regarding the dataset/goals are missing, and the baselines are not really used for materials.
> **Q2** Can the authors compare to materials-focused SOTA methods in the space?
> >> **Reply** Thanks for your suggestion. On the binary classification task of "matbench_mp_is_metal", we provide the updated results as follows:
>
> >>| model  | mean rocauc | mean f1 |
> >>|--------|-------------|---------|
> >>| UniIF  | 0.9617      | 0.9534  |
> >>| CGCNN  | 0.9520      | 0.9462  |
> >>| ALIGNN | 0.9128      | 0.9015  |
> >>| coGN   | 0.9124      | 0.9012  |
> >>| coNGN  | 0.9089      | 0.8972  |
> >>| SchNet | 0.8907      | 0.8765  |
>
> where UniIF outperforms previous baseline methods.
>
>
> **Q1** In the experiments, what exactly does with/without ESM mean? How is ESM used? Why can't the proposed method also become "with ESM"? Why is it valuable to consider "without ESM"? Are models "with ESM" more expensive or what?
> >>**Reply** If ESM is allowed, the model can fintune the pretrained ESM model to refine the designed sequence; Otherwise, the model do not allowed to access the knowledge of ESM. The proposed method can also incooparate with ESM. Without ESM, the algorithm is more efficient, and there is do risk of label leackage to avoid potential concerns.

---

> > ### Author Response · Authors · 2024-08-07
> > **Further clarifications**
> >
> > >> **Reply to W1** In the task of protein design,  GET can achieve 38.3% recovery. This is because GET is designed for protein-molecule interactions learning, and do not responsible for protein design.
> >
> >
> > Here is the citation of CHILI:
> >
> > >> [1] Friis-Jensen, Ulrik, et al. "CHILI: Chemically-Informed Large-scale Inorganic Nanomaterials Dataset for Advancing Graph Machine Learning." arXiv preprint arXiv:2402.13221 (2024).

---

> > > ### Comment · Reviewer_m3n9 · 2024-08-08
> > >
> > > Thank you for the response and efforts for the rebuttal. It is challenging for this reviewer to evaluate the performance of the model on the protein/RNA tasks due to lack of experience in these fields, while the materials task is relatively new and not commonly benchmarked. Since there are some new ideas in the paper and the performance metrics seem to outperform existing works, this reviewer would give the benefit of the doubt for a rating increase from 4 to 5, but with a confidence of 1.

---

> > > > ### Author Response · Authors · 2024-08-08
> > > > **Thank you very much**
> > > >
> > > > Thanks for your review service and raised score!

---

### Official Review · Reviewer_kcUc · 2024-07-09

**Soundness:** 3
**Presentation:** 4
**Contribution:** 4
**Rating:** 5
**Confidence:** 4

**Summary:**

This paper propose a unified framework for inverse folding. Their model is applicable to small molecules, proteins and RNA.
More specifically, they propose a unified frame representation for amino acid, atom and nucleotide. They further propose a GNN-based model to learn the structure information.
Results show that their model can outperform other baselines in protein design, RNA design and material design.


---

I changed my score to 5 after the discussion with other reviewers and AC.

**Strengths:**

(1) The paper is clarified clearly.
(2) The idea of unifying inverse folding for different types of molecules is interesting and reasonable.
(3) Experimental results show their model work good in different design tasks.

**Weaknesses:**

Major points:

1. Some important experimental details are missing. For instance, while the paper proposes a unified inverse folding framework, it's unclear whether the model is trained on a combined dataset from different types of molecules. Specifically, the training sets for RNA design and material design are quite limited. If the model is trained with protein data, it would be beneficial to demonstrate whether incorporating data from other types improves performance.

2. Additionally, if I understand correctly, the paper does not explain the decoder part of the inverse folding model.

Minor points:

1. It would be helpful if the paper included a comparison of perplexity, as related works such as PiFold have used this metric.

2. The performance on protein design looks comparable with other models to me.

**Questions:**

(1) As illustrated in the previous part, do you train the model on three types of molecules?

(2) What decoder model is used?

**Limitations:**

The limitations are not discussed.

---

> ### Author Rebuttal · Authors · 2024-08-07
>
> > **W1** Some important experimental details are missing. For instance, while the paper proposes a unified inverse folding framework, it's unclear whether the model is trained on a combined dataset from different types of molecules. Specifically, the training sets for RNA design and material design are quite limited. If the model is trained with protein data, it would be beneficial to demonstrate whether incorporating data from other types improves performance.
> > **Q1** As illustrated in the previous part, do you train the model on three types of molecules?
> >> **Reply** We train the model on different types of molecules. We have tried to train UniIF in mixed protein and RNA dataset, which achieve slightly lower performance than training on single dataset. We have used all the RNA data recoreded in PDB to train the model, which is actually the largest real data currently available. Maybe we can use data augmentation techiniques in the future.
>
> > **W2** Additionally, if I understand correctly, the paper does not explain the decoder part of the inverse folding model.
> > **Q2** What decoder model is used?
> >> **Reply** A linear layer is served as the decoder to predict 20 amini acids, 4 neuclitides, or 128 atom types.
>
> > **W3** It would be helpful if the paper included a comparison of perplexity, as related works such as PiFold have used this metric.
> >> **Reply** We provide the perplesxity as follows:
>
> | model  | short | medium | long | all  |
> |--------|-------|--------|------|------|
> | PiFold | 5.81  | 4.57   | 3.77 | 4.50 |
> | UniIF  | 5.53  | 4.41   | 3.62 | 4.32 |
>
>
> > **W4** The performance on protein design looks comparable with other models to me.
> >> **Reply** As far as we know, the protein design performance has reached a bottleneck stage where any improvement is difficult.

---

> > ### Comment · Reviewer_kcUc · 2024-08-07
> >
> > Thanks for your reply. I have no further comments.

---

> > > ### Author Response · Authors · 2024-08-08
> > > **Thank you**
> > >
> > > Thanks for your review service and good questions. Have a good day!

---

### Official Review · Reviewer_Aoua · 2024-07-09

**Soundness:** 3
**Presentation:** 3
**Contribution:** 2
**Rating:** 5
**Confidence:** 3

**Summary:**

Previous approaches on molecule inverse folding (IF), which is crucial for drug and material discovery,  focus separately on either macro-molecules or small molecules, leading to the lack of a unified approach for different molecule types. To this end, the paper proposes UniIF to unify the IF from a data and model perspective. Specifically, by introducing a unified block for processing different molecules and a geometric block to capture their 3D interactions, UniIF shows improvements in various tasks across different molecules.

**Strengths:**

- The proposed UniIF is novel and effective for unifying IF across different molecules, including protein, RNA, and small molecules.
- The UniIF can excel in protein, RNA, and material design, which is impressive. Extensive experiments have been conducted to provide justification for the effectiveness of the proposed method.
- The paper is generally well-written, with clear illustrations and tables.

**Weaknesses:**

- The ablation experiments in Table 1 (-GDP), which remove the geometric dot product features, show miner drawbacks in proteins with longer sequences. This makes the geometric interaction extractor mainly capture the interaction of the virtual inter-atoms. The idea for capturing long-range dependency or interaction using virtual blocks has been studied in [1,2,3,4]; the authors could consider providing a discussion on how their proposed techniques differ from these referenced works.
- In addition, the UniIF adopted both node and edge feature extractors as feature augmentation, which is model-agnostic. However, the comparison of all three tasks does not show the potential benefit of introducing such a featureizer.
- The paper could provide further insight into the benefit of employing a unified framework for protein, RNA, and material design. For example, could the UniIF model different molecules in a unified latent space, allowing the researchers to investigate their underlying interactions?

[1] Neural Message Passing for Quantum Chemistry. In ICML, 2017

[2] An analysis of virtual nodes in graph neural networks for link prediction. In LoG, 2022

[3] On the Connection Between MPNN and Graph Transformer. In ICML, 2023.

[4] Neural Atoms: Propagating Long-range Interaction in Molecular Graphs through Efficient Communication Channel. In ICLR, 2024.

**Questions:**

1. Could you provide some discussions or experiments on the effectiveness of the proposed featureizer?
2. Is it possible to train UniIF with mixed molecule datasets?
3. For Equation 7, the SVD decomposition could be time-consuming when scaling the input size. Could the author explain the motivation for such a design and why a simple approach, like "mean," does not work?

**Limitations:**

The proposed UniIF offers a unified framework for learning different molecules, such as protein and RNA. However, it still models different molecules individually, hindering the proposed framework's potential impact.

---

> ### Author Rebuttal · Authors · 2024-08-07
>
> > **W1** The ablation experiments in Table 1 (-GDP), which remove the geometric dot product features, show miner drawbacks in proteins with longer sequences. This makes the geometric interaction extractor mainly capture the interaction of the virtual inter-atoms. The idea for capturing long-range dependency or interaction using virtual blocks has been studied in [1,2,3,4]; the authors could consider providing a discussion on how their proposed techniques differ from these referenced works.
> >> **R1** Thanks for your recommandation. Recent works show that virtual node could help the model to learn long-term dependencies. Our work is different in the virtual frame construction and interaction: we need to construct the local frame for each virtual node and also we use geometric operators to consider the communication between virtual nodes. Other GNN-related works do not consider the 3D geometric information.
>
> > **W2** In addition, the UniIF adopted both node and edge feature extractors as feature augmentation, which is model-agnostic. However, the comparison of all three tasks does not show the potential benefit of introducing such a featureizer.
> > **Q1** Could you provide some discussions or experiments on the effectiveness of the proposed featureizer?
> >> **Reply** Previously, PiFold use a model-related featurizer for protein design and achieves good results. However, such a featurizer is complex and computationally expensive. Our contribution is to simplify the featurizer and mask it model-agnostic, while ensuring competitive performance. If one use PiFold's featurizer, the recovery of protein design can further improve about 0.3%. However, such a improvement is model-related and could not eazily extend to RNA data. With out simplified featurizer, RNA recovery improved much.
>
> > **W3** The paper could provide further insight into the benefit of employing a unified framework for protein, RNA, and material design. For example, could the UniIF model different molecules in a unified latent space, allowing the researchers to investigate their underlying interactions?
> > **Q2** Is it possible to train UniIF with mixed molecule datasets?
> >> **Reply** Yes. We have tried to train UniIF in mixed protein and RNA dataset, which achieve slightly lower performance than training on single dataset. We plan to expend the experiments to protein-RNA-molecule complexes to study the underlying interactions as you suggested.
>
>
> > **Q3** For Equation 7, the SVD decomposition could be time-consuming when scaling the input size. Could the author explain the motivation for such a design and why a simple approach, like "mean," does not work?
> >> **Reply** Simple approach, like "mean", could not obtain a orthogonal frame vectors. We use SVD to get the rotation matrix to serve as orthogonal vector basis.

---

> > ### Comment · Reviewer_Aoua · 2024-08-09
> >
> > Thanks for your response. I retain my score considering the contribution of the submission.

---

> > > ### Author Response · Authors · 2024-08-11
> > > **Thank you**
> > >
> > > Thanks for your review service and appreciation!

---

### Official Review · Reviewer_r3pY · 2024-07-14

**Soundness:** 2
**Presentation:** 3
**Contribution:** 2
**Rating:** 4
**Confidence:** 4

**Summary:**

This paper addresses the challenge of inverse folding, i.e. the design of novel molecules or macromolecules with specific desired 3D structure, with the goal of improving real-world drug and material design. The authors point out that this challenge has been addressed independently for different contexts, such as inverse folding for small molecules vs inverse folding for large macromolecules. They argue that this has resulted in redundant efforts. To address this issue, they propose a unified model for the inverse folding of all molecules. They demonstrate the efficacy of their novel unified approach across three benchmark molecular design tasks involving protein, RNA and material design.

**Strengths:**

The authors make an interesting argument that inverse folding problems in different scientific domains should be addressed by a single approach. They present an elegant approach that unifies approaches across domains. Their results demonstrates improved performance as measured by sequence recovery at benchmark tasks involving protein, RNA and material design. The authors carry out ablation studies that provide some insight into the source of this improved performance.

**Weaknesses:**

While the unified approach is elegant, it is not clear (and the authors do not assert) that the unification is responsible for any performance improvements. However, it is easy for the reader to be left with the impression that the improved results stem from unification. The authors should make it clear that either (i) unification is satisfying but is not connected to performance improvements, or (ii) unification leads to performance improvements and explain clearly the data supporting this.

The recovery metric is confusing, since it is not clear that higher recovery actually corresponds to the ability to design novel molecules. Indeed, the recovery metric penalizes the model for novelty, which seems to contradict the stated goal ('enabling scientists to synthesize novel molecules with the desired structure'). This undermines the claim that the approach presented in the paper 'can benefit multiple domains, such as machine learning, drug design, and material design'. Specifically, rediscovering existing macromolecules does not benefit these domains.

Moreover, even if the metric were aligned with the stated goals, it is not clear to this reviewer that existing inverse folding methods are not good enough to solve any real world application of inverse folding such as the stated goals described in the introduction to this paper of improvements in drug and material design. This leads me to wonder what problem this paper is trying to solve.

While the paper focuses on maintaining the structure of the target molecules (or material), no attempt is made to ensure that the function is maintained. This does not bode well for applications such as drug design, that require that the designed molecule functions correctly.

The paper does not include any wetlab experiments, instead relying on performance at benchmark tasks to demonstrate improved performance. The authors argue that this is the only limitation of this paper, and that wetlab experiments are by definition out of scope for an AI paper. This raises the question of whether improvements on the benchmark tasks and associated metrics used in this paper corresponds to progress on the stated goals of the paper.

**Questions:**

In the absence of wetlab experiments, please could the authors address this limitation by providing data that demonstrates that improved performance at the specific benchmark tasks in inverse folding cited in this paper results in improved performance at novel molecule design. This data can be extracted from previously published work, with appropriate citation.

There have been a series of recent works presenting improvements at inverse folding. These works, including this paper, make lofty claims about the impact that such improvements will have on drug and material design. Please could the authors point to any realized impact on drug and material design resulting from improvements at inverse folding as measured by performance on these benchmark tasks.

If such realized impact is difficult to find, please could the authors identify and address this limitation.

Please could the authors propose strategies to mitigate the risk that further improvements in performance on benchmark tasks, such as the improvements reported in this paper, will not result in realized impact in drug or material design.

Please could the authors propose strategies to mitigate the risk that the metric used in this work penalizes the model for the design of novel molecules, thereby undermining the stated goals.

Please could the authors identify a set of real world applications for inverse folding, describing the problem, how inverse folding will contribute to solving this problem, and why existing inverse folding methods are not good enough to solve each specific real world problem identified (NB inferior performance on benchmark tasks does not demonstrate that existing methods are not good enough).

I note that the number of sequences for CATH 4.3 differs from the referenced papers - please could the authors explain this discrepancy.

For the time split protein tasks, it would be helpful to establish how similar each test sequence is to those used in training, both in terms of proximity in sequence space and in terms of 3D structure. Please then stratify performance as a function of each distance for each held out test set. It would be helpful to also compute these distances for the CATH topology-based split, to provide a watermark for comparison.

**Limitations:**

In addition to the questions above, please could the authors describe limitations of their work that may prevent improvements at inverse design as measured by performance on benchmark tasks using a recovery metric from having impact on real-world challenges such as drug and material design.

---

> ### Author Rebuttal · Authors · 2024-08-07
>
> > **W1:** It is not clear  that the unification is responsible for any performance improvements.
> >> **Reply** The proposed unification method is responsible for performance improvements by using the block-level representation and modules, such as geometric featurizer, interaction operator, and virtual frame.
>
> > The recovery metric is confusing, since it is not clear that higher recovery actually corresponds to the ability to design novel molecules.
> >> **Reply** One might claim that higher recovery reduces diversity without considering similar designability. However, when methods with comparable designability—measured by self-consistent TM (scTM) scores—are considered, the conclusion changes: higher recovery actually increases diversity at the same scTM score level. This is a important finding of ProteinInvBench [1].
> >> [1] Gao, Zhangyang, et al. "Proteininvbench: Benchmarking protein inverse folding on diverse tasks, models, and metrics." NeurIPS (2024).
>
>
>
> > **W2:** Moreover, even if the metric were aligned with the stated goals, it is not clear to this reviewer that existing inverse folding methods are not good enough to solve any real world application.
> > **W3:** The paper lacks wetlab experiments and relies on benchmark performance to demonstrate improvement. Do these benchmark improvements and metrics align with the stated goals?
> > **Q1:** In  the absence  of wetlab experiments, the authors should provide data from published work showing that better benchmark performance in inverse folding leads to improved novel molecule design, with proper citations.
> > **Q2** Can the authors provide examples of improvements in drug and material design resulting from better performance on inverse folding benchmark tasks? If such examples are hard to find, please address this limitation.
> > **Q3:** Please could the authors propose strategies to mitigate the risk that further improvements in performance on benchmark tasks, such as the improvements reported in this paper, will not result in realized impact in drug or material design.
> >> **R3:** The stated goals can be decomposed into two step: computation and wetlab experiments.
> >> Most AI researchers [3,5,6,7] focus on improving  the sequence recovery, which has inspired further biological research. For instance, the well-known ProteinMPNN[4] is largely based on GraphTrans[5]. AI researchers often lack the equipment and funding for wetlab experiments. If you can provide or recommend research opportunities, we would greatly appreciate it.
>
> >> Improving recovery is crucial for wetlab experiments. ProteinMPNN[5] increased recovery from 39% to over 45%, achieving strong wetlab results. LigandMPNN[2] further improved recovery to over 60%, demonstrating the ability to generate small molecules and DNA-binding proteins with high affinity and specificity.
>
> >>  As suggested, AI researchers should collaborate with bio-labs to validate their algorithms. Additionally, evaluating proposed methods on more metrics, such as scTM and diversity, would help mitigate the risks mentioned.
>
> >> [2] Dauparas, Justas, et al. "Atomic context-conditioned protein sequence design using LigandMPNN." Biorxiv (2023).
>
> >> [3] Wu, Fang, and Stan Z. Li. "A hierarchical training paradigm for antibody structure-sequence co-design." NeurIPS (2024).
>
> >> [4] Dauparas, Justas, et al. "Robust deep learning–based protein sequence design using ProteinMPNN." Science (2022).
>
> >> [5] Ingraham, John, et al. "Generative models for graph-based protein design." NeurIPS (2019).
>
> >> [6] Zheng, Zaixiang, et al. "Structure-informed language models are protein designers."ICML, 2023.
>
> >> [7] Hsu, Chloe, et al. "Learning inverse folding from millions of predicted structures." ICML, 2022.
>
>
>
>
> > **W4:** While the paper focuses on maintaining the structure of the target molecules (or material), no attempt is made to ensure that the function is maintained.
> >> **R4:** We appreciate the suggestion; a relevant paper has been published in Nature[6]. We also aim to extend the algorithm to protein functions, such as solubility, but have not found a well-curated dataset for these functions. If you can recommend a suitable dataset, we would be very grateful.
>
> >> [6] Goverde, Casper A., et al. "Computational design of soluble and functional membrane protein analogues." Nature (2024).
>
>
> > **Q4** Could the authors propose strategies to mitigate the risk that the metric used in this work penalizes the model for the design of novel molecules?
> >> **R4** Reducing the temperature can balance recovery and diversity: lower temperatures (temp) increase diversity while reducing recovery. At the same recovery level on CASP15 dataset, UniIF shows higher diversity compared to the baseline, aligning with findings from ProteinInvBench.
>
> |      | ProteinMPNN |      | PiFold |      | UniIF |      |
> |:----:|:------:|:----:|:------:|:----:|:-----:|:----:|
> | temp |   rec  | diversity |   rec  |  diversity |  rec  |  diversity |
> |  0.0 |  0.449 |  0.0 |  0.473 |  0.0 | 0.514 |  0.0 |
> | 0.05 |  0.436 | 0.22 |  0.463 | 0.24 | 0.505 | 0.14 |
> |  0.1 |  0.404 |  0.1 |  0.425 | 0.41 | 0.475 | 0.30 |
>
>
>
> Kindly refer to official comments for more response.

---

> > ### Author Response · Authors · 2024-08-11
> > **Looking forward to your reply**
> >
> > Dear reviewer r3pY,
> >
> > We express our sincere gratitude for your constructive feedback in the initial review. It is our hope that our responses adequately address your concerns. Your expert insights are invaluable to us in our pursuit of elevating the quality of our work. We are fully aware of the demands on your time and deeply appreciate your dedication and expertise throughout this review.
> >
> > We eagerly anticipate your additional comments and are committed to promptly addressing any further concerns.
> >
> > Once again, we extend our heartfelt thanks for your time and effort during the author-reviewer discussion period.
> >
> > Best,
> >
> > Authors

---

> > ### Comment · Reviewer_r3pY · 2024-08-12
> >
> > Thank you for the considered response to my review. I fully agree with the authors that the lack of well curated datasets that would allow AI researchers to address key issues in this domain is a significant issue.
> >
> > I also fully agree with the authors that AI researchers should collaborate with bio-labs to validate their algorithm, and increase their impact on the field - indeed carrying out wetlab experiments is not simply a matter of getting funding and equipment, there is significant technical expertise and experience required, so I would not suggest that AI researchers try setting up experiments from scratch.
> >
> > Navigating collaborations with bio-labs is crucial to achieving significant impact, and I urge these authors to consider reaching out to potential collaborators from other fields, particularly those already equipped with funding, equipment etc.
> >
> > More generally, while I like the idea of a Unified framework across these different tasks, it still remains unclear to me that the unification itself results in improved performance - i.e. that individual tasks benefit from training on the datasets for the other tasks.

---

> ### Author Response · Authors · 2024-08-07
> **Author Rebuttal**
>
> > **Q5** Could the authors identify real-world applications for inverse folding, detailing the problem, how inverse folding could help, and why existing methods fall short for each specific application?
> >> **R4** If the computing benchmark is not recognized and only wetlab experiments are considered, it becomes difficult for us to explain why existing methods fail in applications due to limitations in money, time, equipment, and experience.  Here are some real-world applications of inverse folding:
> - Binding site design[2]: Designing binding sites for small molecules starting from previously characterized designs generated using Rosetta.
> - Membrane protein design[6]: Creating soluble and functional analogues of integral membrane proteins uisng AF2+ProteinMPNN.
> Inverse folding methods are used for sampling new amino acid sequences for a given fold, where improving the recovery is an important goal in these research. By reducing the temperature, one can balance the recovery and diversity. In comparison, our proposed UniIF could balance recovery and diversity better than baselines.
>
> > **Q6** I note that the number of sequences for CATH 4.3 differs from the referenced papers - please could the authors explain this discrepancy.
> >> **R6** Previously, the protein number of CATH4.3 follows the extual description from ESMIF[7].  However, it was recently discovered that they released a dataset with a different number of proteins than what they described. Hence, we correct the description, but actually use the same dataset as in the baseline.
>
> > **Q7** For the time split protein tasks, it would be helpful to establish how similar each test sequence is to those used in training, both in terms of proximity in sequence space and in terms of 3D structure. Please then stratify performance as a function of each distance for each held out test set. It would be helpful to also compute these distances for the CATH topology-based split, to provide a watermark for comparison.
> >> **Q8** There is no way to find the CATH code for novel protein sequences. Following your suggestion, we can provide hirachical performance in terms of sequence and structure similarity using mmseq and foldseek in CASP15 dataset. We found that the sequence similarity positively correlated with the recovery more significantly.
>
> | seq cut |  rec | struct cut |  rec |
> |:-------:|:----:|------------|:----:|
> |   <0.3   | 0.48 | <0.1        | 0.51 |
> |   <0.5   | 0.51 | <0.3        | 0.49 |
> |   <0.8   | 0.52 | <0.8        | 0.52 |

---

> ### Author Response · Authors · 2024-08-13
> **Thanks for your reply!**
>
> Dear reviewer r3pY,
>
> Thank you once again for your reply! We agree with you that AI researchers should consider reaching out to potential collaborators from other fields, particularly those already equipped with funding, equipment etc. We are also looking for such opportunities.
>
> We're glad to hear that you appreciate the unified framework and that we have the chance to clarify your question regarding whether individual tasks benefit from training on datasets for other tasks.
>
> > The answer is no, based on our experiments. But our method already achieves SOTA performance on each single task.
>
> For example, we attempted to train UniIF on a mixed dataset of proteins and RNA, which resulted in slightly lower performance compared to training on a single dataset. However, we would like to emphasize the benefits of the unified framework:
>
> - UniIF can be trained on different molecules within a unified latent space, allowing researchers to explore underlying interactions—similar to what AF3 accomplished. We plan to  extend this in the future.
>
> - UniIF currently achieves state-of-the-art  performance on individual tasks, thanks to the careful design of the model, including  the interaction operator, GNN layer, and virtual frame. To evaluate the effectiveness of UniIF, we also provide the ablation results of RNA design. The performance gain obtained by virtual frame (VFrame) and geometric dot product (GDP) is more significant in RNA dataset. This is because the protein design performance has reached a bottleneck stage where any improvement is difficult. Things will be more clear on RNA dataset.  Here is the RNA ablation results:
>
> >| model      | short | medium | long  | all   |
> >|------------|-------|--------|-------|-------|
> >| w/o VFrame | 44.44 | 49.25  | 37.40 | 45.45 |
> >| w/o EAttn  | 47.89 | 49.00  | 36.71 | 48.28 |
> >| w/o GDP    | 46.88 | 48.00  | 37.19 | 47.06 |
> >| UniIF      | 48.21 | 49.66  | 37.29 | 48.94 |
>
>
>
> - UniIF can serve as a strong baseline that can be further enhanced by incorporating domain-specific knowledge, such as adding secondary structure information for RNA design. That is, individual tasks can be enhanced by adding domain-specific knowledge to UniIF, although UniIF has already achieved SOTA performance.
>
> We sincerely hope this could clarify  your concerns, and we would be happy to answer any additional questions. If we have adequately addressed your concerns, we kindly hope that you consider raising the score to support acceptance. Your comment and suggestion are important and valuable for us. Thank you.
>
> Best,
>
> Authors.

---

> ### Author Response · Authors · 2024-08-13
> **Thanks for your comments!**
>
> Dear Reviewer r3pY,
>
> Thanks for your review. We have tried our best to address your questions and we respectfully thank you for supporting the acceptance of our work. Also, please let us know if you have any further questions. Look forward to further discussions!
>
> Sincerely,
>
> The Authors

---

### Author Response · Authors · 2024-08-14
**Rebuttal Summary**

We thank the reviewers for their insightful and constructive reviews of our manuscript. We are encouraged to hear that the reviewers found the unified framework is **interesting and novel** (Reviewer r3pY,Aoua,kcUc,m3n9), the paper is **well written** (Reviewer Aoua,kcUc) and the proposed **method is effective** (Reviewer r3pY, Aoua, kcUc, m3n9). They think the **method is elegant** (Reviewer r3pY, Aoua, kcUc, m3n9), and extensive experiments have been conducted to provide justification for the **effectiveness** of the proposed method (Reviewer r3pY, Aoua, kcUc, m3n9). During rebuttal, we tried our best to address each reviewer’s concerns point by point, and received major positive comments:
- Reviewer r3pY: I like the idea of a Unified framework across these different tasks but not sure whether individual tasks benefit from training on the datasets for the other tasks.
- Reviewer Aoua: I retain my score (weak accept) considering the contribution of the submission.
- Reviewer kcUc: Thanks for your reply. I have no further comments. (weak accept)
- Reviewer m3n9: There are some new ideas in the paper and the performance metrics seem to outperform existing works.  The reviewer increases the score from 4 to 5 (borderline accept).


We thank all reviewers again for all the efforts that help us improve the manuscript. In case our answers have justifiably addressed your concerns, we respectfully hope that you can increase your score to support the acceptance of our work.

========================================================================


Regarding the remaning question raised by reviewer r3pY, that is "whether individual tasks benefit from training on the datasets for the other tasks", the answer is no. For example, we have tried to train UniIF in mixed protein and RNA dataset, which achieve slightly lower performance than training on single dataset. However, we would like to highlight the merits of the unified framework:

1. One can train UniIF model on different molecules in a unified latent space, allowing the researchers to investigate their underlying interactions, as suggested by reviewer Aoua. Just like what AF3 did, and we will extend this in the future.

2. Currently, UniIF achieves SOTA performance on single tasks, thanks to the carefully model design.

3. One can take UniIF as the strong baseline and incoorpatate their domain-specific knowledge to further enhance the performance, i.e., adding secondary structure information for RNA design.

We believe that incorporating more specific domain knowledge can lead to a stronger model within its domain. However, the unified approach is elegant, and our method already achieves state-of-the-art performance, suggesting that further integrating domain knowledge could unlock even greater potential of UniIF.

---

### Decision · Program_Chairs · 2024-09-25

**Decision:**

Accept (poster)

**Comment:**

This paper considers the problem of inverse folding, which is the design of a chemical (such as a small molecule or a protein) to conform to a given 3D structure. Traditionally, the modelling approach used for this task would depend on whether the object to be designed is a small molecule, protein, or material; here the authors propose the UniIF architecture, which shows competitive results on any of the subdomains of inverse folding.

After mixed initial reviewers, some of the reviewers increased their ratings after author rebuttal, with three in favour of accepting the paper, and one leaning towards rejection. Generally, most reviewers agreed that the paper is interesting, well-written, and relatively novel. They enjoyed the model unification idea, although questioned to what extent it is a useful contribution, given that (as shown by the authors during rebuttal) there seems to be no positive transfer between the domains, and best results are achieved by training UniIF for each task separately. The reviewers also discussed the strength of the empirical results, noting that (a) protein design tasks investigated in this work seem close to saturated, and (b) employed materials design tasks are somewhat uncommon and the results are harder to contextualize due to the choice of baselines. Reviewers continued discussing during the private discussion period, debating whether improvements on top of prior work are clear enough, and whether unification without positive transfer is useful in itself. They noted wetlab results could be a potential way to increase the impact of this work, although they agreed this would not be required for acceptance at NeurIPS. During this phase, two of the positive reviewers decided to lower their scores to borderline accept.

After all of the discussion, the paper ended up with exclusively borderline ratings, albeit not due to lack of opinion on the side of the reviewers, and rather them carefully weighing the pros and cons of this work. On the positive side, it is clear that UniIF is an interesting architecture, and while no advantage from co-training on multiple modalities was observed so far, it could encourage downstream work to investigate this idea further, potentially moving towards a “foundation model” for science. Due to this potential, and generally promising performance, I am willing to lean into the positive aspects of this work and support its acceptance despite borderline ratings. I would encourage the authors to carefully consider the feedback from the reviewers and include relevant discussion in the final version of their paper; in particular, I would recommend highlighting the lack of positive transfer between domains as an avenue for future work.